# Validation of NEDAP Monitoring Technology for Measurements of Feeding, Rumination, Lying, and Standing Behaviors, and Comparison with Visual Observation and Video Recording in Buffaloes

**DOI:** 10.3390/ani12050578

**Published:** 2022-02-25

**Authors:** Ray Adil Quddus, Nisar Ahmad, Anjum Khalique, Jalees Ahmed Bhatti

**Affiliations:** 1Department of Livestock Management, University of Veterinary & Animal Sciences Lahore, Lahore 54000, Pakistan; drraiadil@gmail.com (R.A.Q.); jalees.ahmed@uvas.edu.pk (J.A.B.); 2Department of Animal Nutrition, University of Veterinary & Animal Sciences Lahore, Lahore 54000, Pakistan; akhalique@uvas.edu.pk

**Keywords:** dairy buffaloes, NEDAP technology, validation, video recording, visual observation

## Abstract

**Simple Summary:**

Changes in feeding, rumination, and resting behaviors are directly linked with physiological disturbance and metabolic disorders in animals. Therefore, this study was designed to validate the NEDAP technology for the monitoring of behaviors in buffaloes. The current results showed that feeding, rumination, lying, and standing behaviors were precisely and noninvasively monitored by NEDAP technology than visual observation or video recording in buffaloes. In conclusion, the current finding showed that the NEDAP system can be used for monitoring behavioral variables in buffaloes.

**Abstract:**

The current study aimed to investigate the monitoring behaviors of the NEDAP system in buffaloes, to evaluate the validation, accuracy, and precision over visual observation and video recording. The NEDAP neck and leg tags were attached on the left side of the neck and left front leg of multiparous dairy buffaloes (*n* = 30). The feeding, rumination, lying, and standing behaviors were monitored by the NEDAP system, visual observation, and video recording. The feeding time monitored by NEDAP was 25.2 ± 2.7 higher (*p* < 0.05) than visual observation and video recording. However, the rumination, lying, and standing time was lower (*p* < 0.05) in buffaloes when monitored by the NEDAP technology than by visual observation and video recording. The Pearson correlation between NEDAP technology with visual observation and video recording for feeding, rumination, lying, and standing was 0.91, 0.85, 0.93, and 0.87, respectively. The concordance correlation coefficient between the NEDAP with visual observation and video recording was high for rumination and standing (0.91 for both), while moderate for feeding and lying (0.85 and 0.88, respectively). The Bland–Altman plots were created to determine the association between NEDAP and visual observation and video recording, showing no bias. Therefore, a high level of agreement was found. In conclusion, the current finding showed that the NEDAP system can be used for monitoring feeding, rumination, lying, and standing behaviors in buffaloes. Moreover, these results revealed that the buffalo behavior was monitored precisely using NEDAP technology than visual observation and video recording. This technology will be useful for the diagnosis of diseases.

## 1. Introduction

Cattle, sheep, goats, and buffaloes are the most common species of livestock worldwide. Among those species, buffaloes are the most important and productive animal species in Asian countries and are also called black gold [1]. In Pakistan, buffaloes are raised for milk, meat, and hiding purposes. The production and population of buffaloes are decreasing day by day in Pakistan, as well as all over the world, due to diseases, mismanagement, depression, and other behavioral changes, such as aggressiveness [2]. Therefore, good management at the farm level, disease control, and precaution measurement are very necessary to overcome the issues in the buffalo population.

Good management of buffaloes is necessary for achieving high production, and eating, feeding, rumination, lying, standing, and resting behaviors are the most common variables by which we can evaluate the health status of animals, including buffaloes [3]. In eating or feeding behaviors, the animal’s head moves down for the intake of feed [4]. Rumination is the most important and effective behavior in buffaloes and other livestock animals [5]. In rumination, the size of the feedstuff is reduced in the rumen degradation and takes soluble content from the ration (feed) [6]. Moreover, in rumination behavior, the utilization of forage in animals is highly associated with production, health status, and welfare of animals [7]. Previous studies in the literature showed that changes in feeding and rumination behaviors are directly linked with physiological disturbance [8] and metabolic disorder [9] in animals. Lying and standing both are resting behaviors in animals. An increase in these behaviors is directly correlated with ketosis, milk fever, pain in the abdomen, lameness, or other disorder [10]. Therefore, monitoring these behavioral changes reduces risks and decreases management demands in commercial dairy and fatting farms of cattle and buffaloes.

Behavioral changes in livestock animals were firstly visually observed by a human [11]. This method is usual and reliable for changes in behaviors in commercial dairy farms [12], but this had some limitations [13] (e.g., required a well-trained observer, timing of observation, and difficulty to observe large numbers of animals at a time). To overcome these limitations, another method was video recording that was used in commercial farms [14]. However, there were also some disadvantages to this method, e.g., software problems of the video recorder, being located away from the animals and fixed at one place, and recording no accurate or clear observations [10]. Therefore, to overcome these problems, automated behavior monitoring systems have been used to monitor the feeding, rumination, and resting behaviors of animals.

These systems have advantages over visual observation and video recording. Scientists conducted studies to validate these devices; for instance, in [5], the authors used rumination collars to observe rumination in cattle. Other researchers used ear tag accelerometer 1 [15] and ear tag accelerometers 2 and 3 [16], to evaluate the changes in rumination and feeding activity in Holstein cows. Other researchers used sensors to check behavioral changes in beef cattle [7], sheep, and goats [17], while others checked validation of sensors in free-stall [18] and grazing cattle [8]. These devices potentially have some disadvantages, (e.g., an ear tag system can miscalculate jaw movement with ear movement, and it cannot evaluate lying or resting behaviors) [4]. All of these studies were performed on dairy or beef cattle, but no such type of study has been reported on buffaloes. Therefore, we hypothesized that NEDAP systems have the precise technology to determine the behaviors in buffaloes. The objectives of this study were to investigate the validation of NEDAP neck and leg tags through monitoring of feeding, rumination, lying, and standing behaviors in Nili-Ravi buffaloes, and to compare the behavioral variables monitored by the NEDAP system with visual observation and video recording.

## 2. Materials and Methods

This study was conducted at the Dairy Animals Training and Research Center, B Block, Ravi Campus, UVAS, Pattoki located in Punjab, Pakistan. The methods and procedures of the study were ethically approved by the ethical committee of the University of Veterinary and Animal Sciences, Lahore, Pakistan.

### 2.1. Study Design

The size of experimental animals was determined by a power test as previously described by [19]. Thirty (*n* = 30) lactating Nili-Ravi buffaloes (parity = 2–3) were selected for the study trial. The NEDAP neck and leg tags were attached as instructed by the company, as shown in Figure 1.

### 2.2. Data Collection

The NEDAP-observed data were transmitted from each neck and leg tag of each buffalo on a daily basis. The data were transferred to the NEDAP servers, and the generation of datasheets was evaluated. The data were collected in 2 h observations, four times a day, for 15 days continuously. The buffaloes were observed by 12 video cameras (Panasonic WV BP120, Panasonic, Bracknell, UK), for the observation of feeding, rumination, lying, and standing behaviors. The video cameras were fixed at different key positions in the buffalo shed at a height of 3–4 m. The cameras recorded behaviors on a 24 h/d and weekly basis, and the video was transferred into the observer software (Noldus Information Technology, 2004, Wageningen, The Netherlands). One observer recorded buffalo behaviors visually for 2 h period 4 times a day. Feeding was defined as when buffaloes were at the standing position in front of feedstuff, chewing feedstuff, and ingesting. Rumination was the regurgitation of a bolus chewing and swallowing back [10]. Resting time was separately divided into lying and standing behaviors.

### 2.3. Statistical Analysis

A multivariate general linear model (GLM) was applied to compare the feeding, rumination, lying, and standing behaviors data of NEDAP with visual observation and video recording by using SAS 10.0 version. Person correlation was applied to determine the association between NEDAP, visual observation, and video recording, which was characterized, based on a previous study by [19], as low (0.3–0.5), moderate (0.6–0.7), high (0.8–0.9), and very high (0.9–1.0). To evaluate the accuracy and precision of NEDAP, concordance correlation coefficient (CCC) was applied, and the difference in the standard deviation of NEDAP vs. visual observation and video recording was determined by scale shift and under or over prediction by location shift.

## 3. Results

The descriptive results of NEDAP tags vs. visually observed behaviors and NEDAP vs. video recorded behaviors are represented in Table 1. The average feeding bouts were higher in NEDAP and video recorded than visual observation (human observer). The feeding time was also significantly higher with NEDAP tags than that observed visually and by video recording (25.2 ± 2.7 vs. 24.6 ± 2.6 and 23.1 ± 2.9, respectively; *p* < 0.001), which shows the precision of NEDAP technology over visual observation and video recording in recording buffalo behaviors. The average rumination bouts in buffaloes were significantly higher recorded by NEDAP neck tag than those recorded visually and by video; however, rumination time was significantly higher recorded by video, compared with NEDAP technology and visual observation (19.8 ± 1.7 vs. 18.5 ± 2.1 and 18.6 ± 1.1, respectively; *p* < 0.01). Rumination, feeding, and lying bouts were similar (*p* > 0.05) in all three recording systems.

The average lying bouts variable was significantly (*p* < 0.0001) higher recorded by NEDAP leg tag (20.8 ± 3.6) than that recorded by visual observation (16.4 ± 2.5). The lying time was similar when using NEDAP technology and visual observation but significantly higher in video recording. Standing bouts (resting variable) significantly differed when using NEDAP leg tags, compared with visual observation, but were similar to those recorded by video (3.5 ± 0.1 vs. 2.9 ± 0.1, *p* < 0.0001). The resting time (standing time) in buffaloes was recorded as decreased in NEDAP technology, compared with visual observation and video.

The correlation (r) between NEDAP technology and visual observation for feeding and lying was very high (0.91 and 0.93, respectively; *p* < 0.001), as shown in Table 2. This correlation suggested the high ability of NEDAP neck and leg tags to read feeding and lying behaviors, compared with visual observation. The R value of rumination and standing between NEDAP and visual observation was moderate (0.85 and 0.87, respectively; *p* < 0.01), as shown in Table 2, which suggested a moderate ability for rumination and standing variables. Rumination and standing variables recorded by NEDAP technology strongly correlated with visual observation (CCC ≥ 0.91); however, feeding and lying behaviors recorded moderate correlation (CCC ≥ 0.88), as shown in Table 2. Concordance correlation coefficients show the concurrent measures of correlation, precision, and accuracy. The measure variances in standard deviations between two events, which are shown by the scale shift (μ) being dissimilar from 1.00, and overestimated or underestimated is shown by position shift being dissimilar from 0.00. Scale and location shifts were used to estimate the bias correction factor (**C_b_**) and were multiplied by the estimated Pearson correlation coefficient to determine CCC.

The difference between NEDAP neck and leg tag reading and visual observations for feeding, rumination, and lying is shown in Figure 2. The Bland–Altman plot was created to evaluate the difference between the reading of NEDAP technology and human observation for Nili-Ravi buffaloes. This plot indicated whether the reading of NEDAP technology for feeding, rumination, and lying variables was overestimating (positive bias) or underestimating (negative bias). The results of mean difference for feeding and rumination were 6.3 and 9.6 min, respectively, and bias correlation (**C_b_**) was 1.00 and 0.97, with the mean difference near 0, and the mean difference for lying was −10.3 min. The results of mean difference indicate that the technology was accurate for feeding and rumination behaviors when compared with visual observation, but the estimates of lying or resting were less accurate.

## 4. Discussion

Livestock animals are the backbone of the agricultural economy worldwide. To increase the production of these animals, normal health status and disease control are very necessary [20]. Dairy cattle show less feeding behavior and ruminate less when diagnosed with any metabolic disorder [18] or digestive diseases [21], as well as at the time of calving [5]. Therefore, precise monitoring technologies are useful to monitor the behaviors of animals (feeding, rumination, lying, resting, and inactive variables of animals) [7]. Moreover, different scientists used different technologies to monitor the normal behaviors of animals to differentiate normal [12] and abnormal conditions of animals [6]. All previous studies were conducted on dairy [3] and beef cattle [22], or Holstein or crossbreed [10], with different nutritional conditions. To the best of our knowledge, this was the first validation study, in which NEDAP neck and leg tag loggers were used to monitor the feeding, rumination, lying, and standing behaviors in dairy buffaloes. In this study, we also compared NEDAP technology with visual observation and video recording. These findings are also important for improving the production and diagnoses of diseased status in buffaloes.

The results of the current study showed that average feeding bouts and time were monitored as higher using the NEDAP technology than those observed by visual observation; however, rumination time monitored by NEDAP was in agreement with visual observation. When comparing the current study’s findings with previous technologies used by various scientists, we found that NEDAP technology for monitoring feeding and rumination behaviors in buffaloes is very precise and more validated than other available technologies. In previous reports, accelerometer ear tag 1 was validated by scientists [4] in free-stall housed dairy cows; in [7], the authors reported behavioral-monitoring collars in beef cows and found that ear tag 1 and BMC had high precision for rumination and feeding but lower than that the precision of NEDAP technology. However, the authors of [12] used novel sensors in dairy cattle, while in [20] and [15], researchers used monitoring technology ear tags 2 and 3 in dairy cattle. They all found much lower precision and correlation in feeding and rumination behaviors than our current finding (NEDAP technology). The difference in the results of precision in technologies may be due to species difference or that those studies used free-stall housed conditions and grazing feeding, but in our study, we monitored behaviors in pen housing system and feeding. The distance from monitoring of behaviors much affected the validations and precision of technologies [18]. The study in reference [13] used a microphone collar, and reference [11] used a noseband in dairy cattle; both reported similar validation in terms of feeding and rumination behaviors, compared with using the NEDAP technology either in Holstein cows [10] or buffaloes, as in the current study. The similarity in these findings may be due to a similar nutritional ratio or condition of data collection. The feeding and rumination time by monitoring of NEDAP was higher in the current study than those indicated in [10] in Holstein cows. The higher feeding and rumination time with the use of the same technology in both studies may be due to species differences or differences in the nutritional ratio or position of NEDAP attachment.

The monitoring of behavioral changes in cattle and buffaloes is very necessary. The animals resting behavior has been monitored by various technologies. In [23], the authors used a 3D pedometer to assess the active motion and resting time of dairy cows in a tie-stall housed system, while in another study [15], the authors used RumiWatch to monitor the locomotion of dairy cows in grazing land. Both found that cows resting times were similar using monitoring technology or visual observation. The resting behavior (lying time) was similar when monitored by NEDAP or visual observation; however, standing time was less when monitored by NEDAP than that by visual observation. The difference between NEDAP and visual observation may be due to inaccurate observation of human observers or inaccurate timing of human observers. In a previous report, the resting behavior was measured by a behavioral-monitoring collar, and the results were in agreement with visual observation in cattle [18]. Another scientist [6] also observed similar results on lying and inactive behaviors in cows. It is not surprising that resting behaviors (lying and standing) had similar results when monitored by visual observation and monitoring technology, as there is no movement in resting behaviors [17]. The monitoring of lying behavior is very important for the diagnosis of disease conditions such as ketosis [21], or when animals are lethargic [12] or have depression [3].

A direct correlation is present between behavioral change in animals and any type of disease [1]. Therefore, in the current study, the correlation of NEDAP tags and visual observation was determined based on feeding, rumination, lying, and standing variables. In the current study, we found a high correlation of feeding and lying behaviors (0.9; *p* < 0.001) between NEDAP and visual observation; however, a moderate correlation was found in rumination and standing behaviors. A similar finding was observed in [5], using rumination collars, in [4], using ear attachment sensor, in [8], using Rami watch system on grazing dairy cows, and in [24], using monitoring technology on free-stall housed cattle. In the current study, the CCC correlation between NEDAP and visual observation was moderate for feeding and lying behaviors but high for rumination and standing behaviors. Our findings are in agreement with those reported in previous studies—namely, in [23], using a 3D pedometer sensor in tie-stall cows; in [20], using monitoring technology on dairy Holstein cattle, in [10], using NEDAP technology in dairy cows. However, little change was present in the CCC correlation of technology and visual observation in the finding on grazing cattle [15], and free-stall cattle, sheep, and goats [17]. These findings suggested that validation and precision can be changed by using technologies on different housed systems animals [18] and by the health status of animals [9].

The findings of the current study for behavioral changes that were monitored by NEDAP vs. visual observation and video recording suggested a high precision of NEDAP neck and leg tags in buffaloes, based on high bias correlation (**C_b_** 1.00 and 0.97), with the mean difference near 0 for feeding and rumination. In [20], the authors found high bias correlations for feeding and rumination behaviors, which were monitored by accelerometer ear tags. Another researcher [13] used a microphone collar in grazing cows and reported a high bias correlation, with 95% interval agreement of the Bland–Altman plot. In [11], the authors used a nose pressure band, while the authors of [22] used behavioral monitoring collars in beef cows; both reported high Bland–Altman plot agreement, with a high bias correlation. All these studies showed high accuracy of monitoring devices on feeding, rumination, and resting activities in dairy and beef cattle. All these reports were performed on cattle but no such types of monitoring systems have been used for buffaloes to detect the metabolic disorders earlier and improve the health status of buffaloes.

In our view, previously, different sensors or monitoring systems were used to monitor behaviors in dairy and beef cattle. All these sensors monitored feeding [14] or eating [8] considerably accurately and sufficiently, and they are reliable. For resting ear [6], neck [16], and head, collars are not sufficient due to the fact that their position can be misplaced during feeding, etc. While it is true that collars can be misplaced, ear tag accelerometers cannot [15,18]. They stated that human observation is also a reliable source and method to detect feeding, rumination, eating, and resting behavioral changes in dairy and beef cattle [10]. There is a limitation in this method, i.e., the limitation of a well-trained observer and accurate timing by observer [4]. The findings of the current study eliminate such types of problems in monitoring technology, and these findings also improve our understanding of the early response against abnormal changes.

## 5. Conclusions

To the best of our knowledge, this is the first study to validate the precision of NEDAP neck and leg tags monitoring for feeding, rumination, lying, and standing behaviors in buffaloes. The feeding, rumination, lying, and standing behavioral changes monitored by NEDAP tags were highly precise and acceptable, compared with visual observation and video recording. The results of our study suggest that the NEDAP technology can be used in commercial dairy buffalo farms to monitor behavioral changes for scientific, health, and management purposes. Further research is needed to evaluate the technology regarding diseases diagnosis in buffaloes.

## Figures and Tables

**Figure 1 animals-12-00578-f001:**
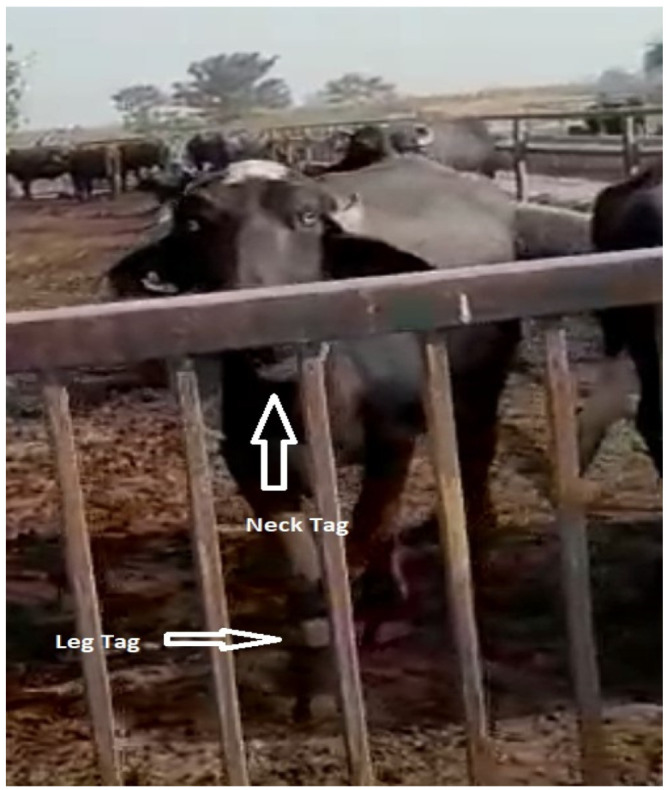
The black arrow shows the NEDAP neck tag, while the red arrow shows the NEDAP leg tag.

**Figure 2 animals-12-00578-f002:**
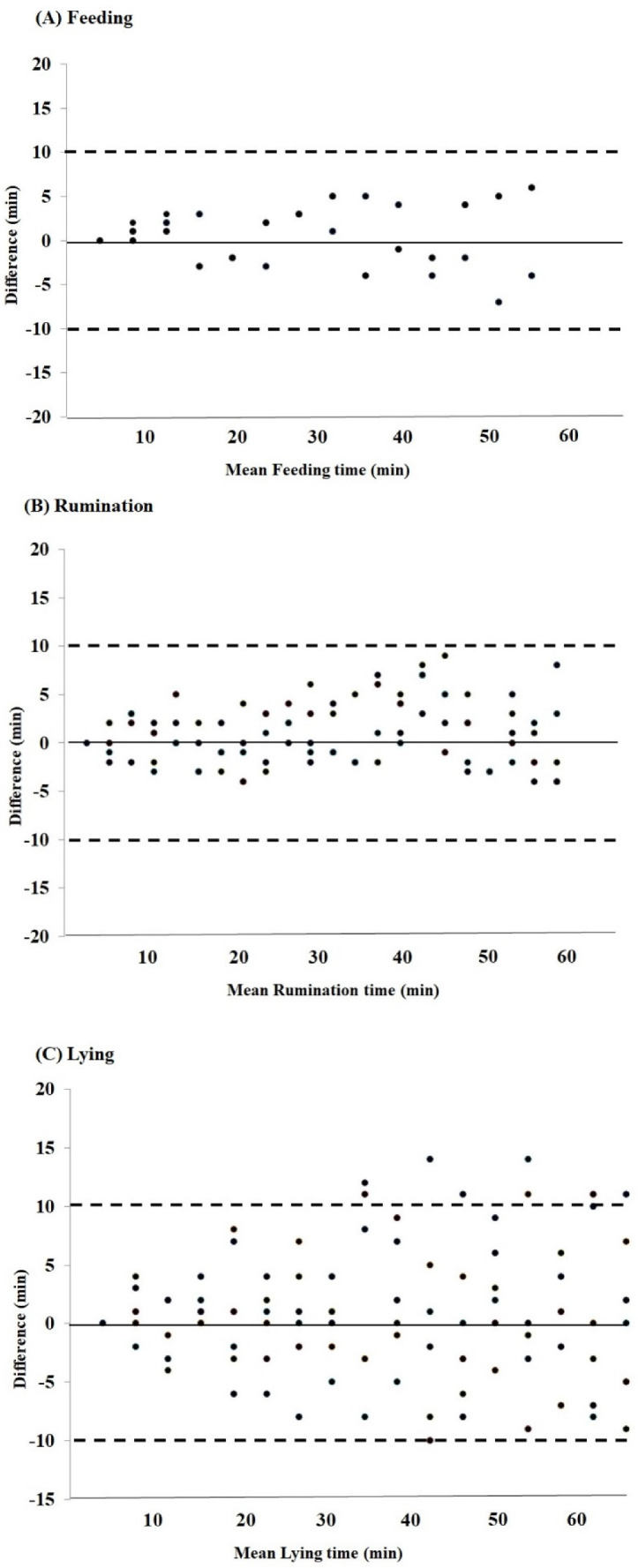
Bland-Altman plots of (**A**) feeding, (**B**) rumination, and (**C**) lying; the results from a validation study between NEDAP tags and visual observation measured behaviors in Nili–Ravi buffaloes for 120 hourly observations.

**Table 1 animals-12-00578-t001:** Descriptive statistics of a validation study for NEDAP leg and neck tag or visual observation or video recorded behaviors in Nili-Ravi buffalos (*n* = 120 h observations).

Variables		NEDAP			Visual Observation			Video		
		Mean ± SD	Min	Max	Mean ± SD	Min	Max	Mean ± SD	Min	Max
Feeding	FB	2.1 ± 1.8	0	8	2.3 ± 0.6	0	6	2.2 ± 0.4	0	7
	AFB	10.8 ± 3.7 ^a^	0	84.0	8.5 ± 1.1 ^b^	0	42.1	10.2 ± 1.6 ^a^	0	60
	FT	25.2 ± 2.7 ^a^	0	84.0	24.6 ± 2.6 ^b^	0	75.5	23.1 ± 2.9 ^c^	0	76.6
Rumination	RB	1.8 ± 1.5	0	7	1.9 ± 0.5	0	6	1.9 ± 0.7	0	6
	ARB	7.9 ± 1.1 ^a^	0	30.8	6.8 ± 1.2 ^c^	0	20.5	7.1 ± 1.1 ^b^	0	23.1
	RT	18.5 ± 2.1 ^b^	0	90.4	18.6 ± 1.1 ^b^	0	85.2	19.8 ± 1.7 ^a^	0	71
Lying	LB	1 ± 0.4	0	3	1.4 ± 0.1	0	3	1.5 ± 0.5	0	3
	ALB	20.8 ± 3.6 ^a^	0	89.4	16.4 ± 2.5 ^b^	0	89.7	19.2 ± 2.5 ^ab^	0	89
	LT	26.9 ± 2.5 ^b^	0	89.4	26.4 ± 2.8 ^b^	0	89.7	27.5 ± 2.2 ^a^	0	89.3
Standing	SB	3.5 ± 0.1 ^a^	0	4	2.9 ± 0.1 ^b^	0	5	3.5 ± 0.7 ^a^	0	4
	ASB	17.6 ± 1.4	0	36.2	18.2 ± 2.3	0	32	17.9 ± 1.4	0	29.8
	ST	53.8 ± 2.5 ^c^	1	77.5	57.5 ± 7.8 ^a^	1	90	55.4 ± 4.3 ^b^	0	89.6

FB: feeding bouts (number); AFB: average feeding bouts (min); FT: feeding time (min); RB: rumination bouts (number); ARB: average rumination bouts (min); RT: rumination time (min); LB: lying bouts (number); ALB: average lying bouts (min); LT: lying time (min); SB: standing bouts (number); ASB: average standing bouts (min); ST: standing time (min). Superscripts on different means within row differ significantly at *p* ≤ 0.05.

**Table 2 animals-12-00578-t002:** Correlation, accuracy, and precision of a validation study between NEDAP variables, visual observation, and video recorded behaviors in Nili-Ravi buffalos (*n* = 120 h observations).

Variables	Pearson Correlation (r)	*p* Value	Bias Correlation (C_b_)	Location Shift (V)	Scale Shift (µ)	CCC (95% Cl)
Feeding	0.91	0.001	1.00	−0.003	0.98	0.85 (0.81 0.92)
Rumination	0.85	0.01	0.97	0.002	1.01	0.91 (0.88 0.93)
Lying	0.93	0.001	0.99	0.16	0.91	0.88 (0.91 0.94)
Standing	0.87	0.01	1.00	0.08	0.89	0.91 (0.88 0.93)

## Data Availability

Data will be provided on demand if required.

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
