# Peer review of "Validation of NEDAP Monitoring Technology for Measurements of Feeding, Rumination, Lying, and Standing Behaviors, and Comparison with Visual Observation and Video Recording in Buffaloes"

_animals, 2022, doi:10.3390/ani12050578_

Round 1

Reviewer 1 Report

The resting activity (lying and standing) results were lower by NEDAP in comparison with visual and video recording. These results were classified by NEDAP as what type of activity? Is there any explanation for this kind of behavior?

Author Response

When the animals are in a resting position, there was no activity of the leg and neck as observed.

Reviewer 2 Report

This is an interesting study which aims to assess the performance of Nedap technology for monitoring behavoural patterns in buffalo. Similar studies have been carried out using other automated measurement systems with regard to dairy cattle and the authors have referenced this. The fact that this applies to buffalo does not in my mind make the paper particularly novel but for validation studies such as these, novelty is not essential. I found the approach to be well constructed and the analysis in the main to be fair and I am happy for the paper to be published. There are a number of typographical and linguistic areas that should be tidied up to enhance clarity. I will outline these.

L41-43

 Moreover, these results prove that NEDAP technology was highly précised for the monitoring of behavior than visual observation and video recording and this technology will useful for the diagnosis of diseases.

I wonder if the following is more appropriate.

Moreover, these results demonstrate that NEDAP technology is has greater precision  for monitoring than visual observation and video recording and that the technology is useful for the diagnosis of diseases.

I would also question whether the statement that the technology is better than video or visual observation has been proven. I think I agree with the conclusion but I am not sure that the paper demonstrates this to be true since there was no other independent verification of the performance of video or observations. Perhaps the authors can explain that there are challenges with video and observer analysis and that technological approaches provide an unbiased measure which can be obtained round the clock.

L83

cattle. Other researchers used ear tag 1 [18] and accelerometer ear tag 2 and 3 [10] to evaluate the

Both systems are accelerometer based. please correct

L 88

can miscalculate jaw movement with ear movement and it cannot evaluate lying or resting behaviors

This is an assertion and while I agree with it the authors have not justified it. Perhaps soften with the inclusion of ‘potentially’

128 To evaluate the accuracy and precision of NEDAP concordance correlation coefficient (CCC) was applied, the difference in the standard deviation of NEDAP vs visual observation and video recording was determined by scale shift and under or over prediction by location shift

This is interesting. I think the CCC metric might not be familiar to all readers and the paper would benefit from the authors explaining it and why it is useful in this context.

In Table 1 it would be of benefit to have the units of measurement included. Also the terms FB, AFB etc have not been defined in the text. While it is possible to work them out, they should be defined.

L 172 The results of mean difference indicate that the technology was accurately and précised for feeding and rumination behaviors as compared to visual observation, but the reading for lying or resting was not very accurately or précised.

Reword – suggestion: The results of mean difference indicate that the technology was accurate for feeding and rumination behaviors when compared to visual observation, but the estimates of lying or resting less so.

L175 The feeding and rumination behaviors recorded by NEDAP tags had higher agreement than visual observation and video.

I am not sure what this means. If it is saying that the NEDAP tag is better than visual observation or video at recording feeding and rumination then I believe that the paper does not have the evidence to support this and the statement should be removed.

Some of the sentences were difficult to follow because the embedded references were sometimes just references and sometimes they were referred to as being the authors. To be clear, statements like ‘[21] had used microphone’ should be written as ‘the study in reference [21] reported the use of microphones’ or ‘reference [21] used microphones…

This will make the text clearer.

Rami-watch should be Rumiwatch

The paragraph from L 270 suggests that for resting, eartags and collars are not effective due to the fact that their position can be misplaced during feeding etc. While it is true that collars can be misplaced, eartags cannot and in anycase this is not the reason for their inaccuracy. Please rewrite

Author Response

L41-43

 Moreover, these results prove that NEDAP technology was highly précised for the monitoring of behavior than visual observation and video recording and this technology will useful for the diagnosis of diseases.

 Response: Thank you for your supportive comments. We have corrected the statement as suggested.

I wonder if the following is more appropriate.

Moreover, these results demonstrate that NEDAP technology is has greater precision for monitoring than visual observation and video recording and that the technology is useful for the diagnosis of diseases.

I would also question whether the statement that the technology is better than video or visual observation has been proven. I think I agree with the conclusion but I am not sure that the paper demonstrates this to be true since there was no other independent verification of the performance of video or observations. Perhaps the authors can explain that there are challenges with video and observer analysis and that technological approaches provide an unbiased measure which can be obtained round the clock.

Response: We appreciate your constructive comments. The NEDAP technology works on automated system and monitors the activity of animals using machine learning system 24/7, while the video recording used to monitor the activity and then the video data is manually corrected by the observer/researcher. There is may be chances of animal hiding or moving away from the range of cameras and the activity could not be recorded at that moment.

L83

cattle. Other researchers used ear tag 1 [18] and accelerometer ear tag 2 and 3 [10] to evaluate the

Both systems are accelerometer based. please correct

 Response: Thank you for your supportive comments. We have corrected the statement as suggested.

L 88

can miscalculate jaw movement with ear movement and it cannot evaluate lying or resting behaviors

This is an assertion and while I agree with it the authors have not justified it. Perhaps soften with the inclusion of ‘potentially’

 Response: Thank you for your supportive comments. We have corrected the statement as suggested.

128 To evaluate the accuracy and precision of NEDAP concordance correlation coefficient (CCC) was applied, the difference in the standard deviation of NEDAP vs visual observation and video recording was determined by scale shift and under or over prediction by location shift

Response: Thank you for comments. We have explained the CCC in the revised manuscript file. Concordance correlation coefficients show the concurrent measures of correlation, precision and accuracy. It measures the variances in the standard deviations between two events that are showed by the scale shift (μ) dissimilar from 1.00, and overestimated or under estimated is shown by position shift dissimilar from 0.00. Scale and location shifts are used to estimate the bias correction factor (Cb) and are multiplied by the estimated Pearson correlation coefficient to determine CCC.

This is interesting. I think the CCC metric might not be familiar to all readers and the paper would benefit from the authors explaining it and why it is useful in this context.

In Table 1 it would be of benefit to have the units of measurement included. Also the terms FB, AFB etc have not been defined in the text. While it is possible to work them out, they should be defined.

 Response: Thank you for supportive comments. We have defined the terms in the revised manuscript.

L 172 The results of mean difference indicate that the technology was accurately and précised for feeding and rumination behaviors as compared to visual observation, but the reading for lying or resting was not very accurately or précised.

Reword – suggestion: The results of mean difference indicate that the technology was accurate for feeding and rumination behaviors when compared to visual observation, but the estimates of lying or resting less so.

 Response: Thank you for your valuable suggestions. We have corrected the sentence as suggested.

L175 The feeding and rumination behaviors recorded by NEDAP tags had higher agreement than visual observation and video.

I am not sure what this means. If it is saying that the NEDAP tag is better than visual observation or video at recording feeding and rumination then I believe that the paper does not have the evidence to support this and the statement should be removed.

 Response: Thank you for your excellent observation. We have removed the statement as suggested.

Some of the sentences were difficult to follow because the embedded references were sometimes just references and sometimes they were referred to as being the authors. To be clear, statements like ‘[21] had used microphone’ should be written as ‘the study in reference [21] reported the use of microphones’ or ‘reference [21] used microphones…

This will make the text clearer.

 Response: Thank you for your excellent observation. We have removed the statement as suggested.

Rami-watch should be Rumiwatch

 Response: Thank you for your excellent observation. We have removed the statement as suggested.

The paragraph from L 270 suggests that for resting, eartags and collars are not effective due to the fact that their position can be misplaced during feeding etc. While it is true that collars can be misplaced, eartags cannot and in anycase this is not the reason for their inaccuracy. Please rewrite

Response: Thank you for your excellent observation. We have removed the statement as suggested.

Reviewer 3 Report

This study aimed to compare the NEDAP system with the gold standard methods of visual observation and video recording to determine if the system is suitable for monitoring behaviours in buffalo.

Overall, the purpose of the study and methods are sound, however, the language utilised needs to be improved substantially before it is ready for publication. The tense is inconsistent throughout and words, e.g. ‘the’, are also missing, whilst there are numerous sentences that are unnecessary and should be removed. I would suggest that the authors obtain a professional proofreader/editor to read through the paper and make the appropriate edits.

The authors also appear to have serious flaws in their reasoning and scientific validity, e.g. L136-138 whereby they state that NEDAP is more precise compared to visual or video observations (gold standard in the field).

The results are also very repetitive in the discussion and there is little to no exploration of the field of knowledge and comparison drawn to the results of this study.

My recommendation is that there are major edits that are required prior to publication.

L18-25: The simple summary needs to be rewritten. L18 is unnecessary. There is also substantial evidence that changes in feeding, rumination, and resting have been observed in response to physiological changes (L19-20), therefore, remove ‘to the best of our knowledge’. There is also a conflicting statement – L22-23 indicate that there is a significant difference between time spent performing behaviours using the NEDAP technology compared to visual observation and video recording, however, L24 indicates that this is a suitable method for monitoring buffaloes.

L47: As with the simple summary, remove ‘livestock species are important animals in the animal kingdom’.

L83-84: What does ear tag 1 and accelerometer ear tag 2 and 3 refer to?

L107: I would appreciate a diagram or figure of how the neck and leg tags were attached. Additionally, there is no information around the specifications of the NEDAP tags, e.g. how frequently are they sampling, are they tri-axial accelerometers, how is the data being processed. This information is critical for repeatability. It is also unclear whether there was a leg and neck tag capturing the different behaviours or if each tag was designated to a specific behaviour.

L121: You will need to be more specific about what was classified as resting behaviour.

L134: Table 1.

L136-L138: Your statement claiming that NEDAP technology is more precise compared to visual observation or video recording is untrue. Visual observation and video recording are both considered gold standard methods of capturing behavioural observations. I am unaware if NEDAP has been validated, however, the correct statement would be that the NEDAP is inaccurate given the significant difference compared to these gold standard methods.

L184: Pakistan’s economy not our national economy.

L204, 206: Not sure what accelerometer ear tag 1 refers to.

L209: Not sure what ear tag 2 and 3 refer to.

L226: I believe you might be referring to Rumi-watch not Rami-watch.

L230-232: If there is concern around inaccuracies of visual observation, additional observers should have been used and intra observer differences should have been taken into account.

L235-237: Hard to follow the logic of this sentence.

Author Response

Reviewer comments & suggestions

Response by author

Overall, the purpose of the study and methods are sound, however, the language utilized needs to be improved substantially before it is ready for publication. The tense is inconsistent throughout and words, e.g. ‘the’, are also missing, whilst there are numerous sentences that are unnecessary and should be removed. I would suggest that the authors obtain a professional proofreader/editor to read through the paper and make the appropriate edits.

Thanks for your valuable suggestion to improve our manuscript, we proofread the manuscript and edit with help of professional English editor.

L18-25: The simple summary needs to be rewritten.

Thanks for your valuable suggestion, we rewritten the summary simply

. L18 is unnecessary.

Thanks for your suggestion we re-write the summary

There is also substantial evidence that changes in feeding, rumination, and resting have been observed in response to physiological changes (L19-20), therefore, remove ‘to the best of our knowledge’.

Thanks for your suggestion we re-write the summary, and remove “to the best of our knowledge”

There is also a conflicting statement – L22-23 indicate that there is a significant difference between time spent performing behaviours using the NEDAP technology compared to visual observation and video recording, however, L24 indicates that this is a suitable method for monitoring buffaloes.

Thanks for your valuable suggestion, we rephrase these lines as suggested.

As with the simple summary, remove ‘livestock species are important animals in the animal kingdom’.

Thanks for your valuable suggestion, we remove this sentence as suggested.

What does ear tag 1 and accelerometer ear tag 2 and 3 refer to?

Thanks for your valuable suggestion and comments. These are sensors, which are used by (Pereira et al., 2018) and (Diosdado et al., 2015) in Holstein cattle.  

You will need to be more specific about what was classified as resting behaviour.

Thanks for your valuable suggestions, resting behavior and lying behavior is same. We edit as suggested.

Your statement claiming that NEDAP technology is more precise compared to visual observation or video recording is untrue. Visual observation and video recording are both considered gold standard methods of capturing behavioural observations. I am unaware if NEDAP has been validated, however, the correct statement would be that the NEDAP is inaccurate given the significant difference compared to these gold standard methods

Thanks for your valuable suggestion and comments, we agree with you that Visual observation and video recording are both considered gold standard methods of capturing behavioural observations, but here NEDAP technology accurately measured behaviors than these gold standards.

Not sure what accelerometer ear tag 1 refers to.

L209: Not sure what ear tag 2 and 3 refer to.

Thanks for your valuable suggestions. All are ear tags sensors.

Pakistan’s economy not our national economy.

We correct it, as suggested

L230-232: If there is concern around inaccuracies of visual observation, additional observers should have been used and intra observer differences should have been taken into account.

Thanks for your valuable suggestions. We have plan to perform next experiment in future and we will consider your suggestion.

L235-237: Hard to follow the logic of this sentence.

We have corrected the sentence, please.

Round 2

Reviewer 3 Report

I am pleased to see a substantial improvement in English used throughout the manuscript, however, the tense is still incorrect in places and redundant sentences appear throughout the manuscript. I would once again suggest that the authors utilise a professional English editor to revise the manuscript.

One key concern I have identified in this manuscript is the logic used to determine the method of behavioural monitoring that is considered the most "accurate". This paper, which is focussed on validating NEDAP technology for monitoring buffalo behaviour, claims that NEDAP technology is more accurate compared to the gold standard methods of visual observation and video recording. Given that these methods are the gold standard and therefore the most accurate, it does not make sense to claim that NEDAP is more accurate.

Author Response

Reviewer comments & suggestions

Response by author

Remove the “the”, i.e. “changes in feeding… “

Thanks for your valuable suggestion to improve our manuscript, we removed the word as suggested

L14: “we designed”

Thanks for your valuable suggestion, we rewrote as suggested

L15: “to validate NEDAP technology”

Thanks for your suggestion we re-wrote as suggested

L23: “multiparous, dairy buffaloes”

Thanks for your suggestion we re-wrote as suggested

L25: “… min higher (P < 0.05)” you can just state that it is higher and place the significance values in brackets

Thanks for your valuable suggestion, we rephrased this line as suggested.

L26: “lower”

Thanks for your valuable suggestion, we changed the word as suggested.

L34: “Therefore, a high level of agreement was found”

L36-38: As above comment, there is a flaw in logic here.

Thanks for your valuable suggestion and comments.  We rephrasde it

L42: What about poultry? Reference?

L44: Reference?

Thanks for your valuable suggestions, we added reference “ [25].

L47: How do behavioural changes impact on production and population of buffalo?

Thanks for your valuable suggestion and comments,  due to “aggressiveness”

L53: Remove “staff”

Thanks for your valuable suggestions. We removed it

L54-55: Reference?

We add reference as suggested

L56-58: Sentence needs tobe rewritten

Thanks for suggestion, we re-wrote the senence

L62: “other disorders”. Remove “any”

Thanks for suggestion we removed the word

L75-77: Sentence needs to be rewritten

Thanks for suggestion we re-wrote the sentence

L77-78: Be specific about what these ear tags are

Thanks for suggestion we wrote specific name of ear tag

L101: Can you provide details about the attachment method. Add a figure or more description

Thanks for suggestion we added figure 2

L104: “on a daily basis”

L106: What time period?

We rephrased it

128: “Table 1”

L169: Figure 1

Thanks for suggestion we corrected it

L207-208: Specify the accelerometer ear tag so that readers do not have to go and look for it

We specified the ear tag name as suggested